# Probiotics: Symbiotic Relationship with the Animal Host

**DOI:** 10.3390/ani12060719

**Published:** 2022-03-12

**Authors:** Elvia Guadalupe Melara, Mavir Carolina Avellaneda, Manuel Valdivié, Yaneisy García-Hernández, Roisbel Aroche, Yordan Martínez

**Affiliations:** 1Master Program in Sustainable Tropical Agriculture, Graduate Department, Zamorano University, P.O. Box 93, Valle de Yeguare, San Antonio de Oriente 11101, Honduras; elvia.melara.m2021@alumni.zamorano.edu; 2Plant Pathology, Diagnosis and Molecular Research Lab, Agricultural Sciences and Production Department, Zamorano University, P.O. Box 93, San Antonio de Oriente 11101, Honduras; cavellaneda@zamorano.edu; 3National Center for Laboratory Animal Production, P.O. Box 6240, Santiago de las Vegas, Rancho Boyeros, Havana 10900, Cuba; mvaldivie@ica.co.cu; 4Departamento de Animales Monogástricos, Instituto de Ciencia Animal, Carretera Central km 47 ½, San José de las Lajas 32700, Cuba; yaneisyg@ica.co.cu; 5Department of Animal Husbandry, Faculty of Agricultural Sciences, University of Granma, Bayamo 85100, Cuba; rarocheg@udg.co.cu; 6Poultry Research and Teaching Center, Agricultural Science and Production Department, Zamorano University, P.O. Box 93, Valle de Yeguare, San Antonio de Oriente 11101, Honduras

**Keywords:** animal host, animal production, beneficial microorganism, gut health

## Abstract

**Simple Summary:**

Intestinal health directly influences the profitability of animal production, and so growth-promoting antibiotics have been used in the feed or drinking water to reduce the impact of enteric diseases and improve production parameters. However, these have generated long-term bacterial resistance. In the search for natural alternatives to antibiotics, various probiotic strains have been developed to improve intestinal health and biological indicators in farm animals, which is important to provide the consumer with safe food. This review describes the main probiotic bacteria and yeasts, their in vitro properties and their impact on the antioxidant capacity and intestinal environment of animals. Furthermore, this review outlines the role of probiotics in apparently healthy ruminants, pigs and poultry, including animals with digestive diseases.

**Abstract:**

Antibiotic growth-promoters in animal feeding are known to generate bacterial resistance on commercial farms and have proven deleterious effects on human health. This review addresses the effects of probiotics and their symbiotic relationship with the animal host as a viable alternative for producing healthy meat, eggs, and milk at present and in the future. Probiotics can tolerate the conditions of the gastrointestinal tract, such as the gastric acid, pH and bile salts, to exert beneficial effects on the host. They (probiotics) may also have a beneficial effect on productivity, health and wellbeing in different parameters of animal performance. Probiotics stimulate the native microbiota (microbes that are present in their place of origin) and production of short-chain fatty acids, with proven effects such as antimicrobial, hypocholesterolemic and immunomodulatory effects, resulting in better intestinal health, nutrient absorption capacity and productive responses in ruminant and non-ruminant animals. These beneficial effects of probiotics are specific to each microbial strain; therefore, the isolation and identification of beneficial microorganisms, as well as in vitro and in vivo testing in different categories of farm animals, will guarantee their efficacy, replicability and sustainability in the current production systems.

## 1. Introduction

It is widely known that the microbiota of the gastrointestinal tract (GIT) is very diverse and dense, made up of bacteria, archaea, viruses and protozoa. Bacteria are the dominant group at GIT with at least 500 different bacterial species of the *Bacteroidaceae* family, among them *Bacteroids* spp., *Bifidobacterium* spp., *Eubacterium* spp., *Clostridium* spp. and *Lactobacillus* spp. [1,2]. Specifically, these bacteria maintain a symbiotic relationship with the host intestinal mucosa and coexist within GIT [3]. The composition and concentration of this microbiota vary depending on the physiological characteristics of the different parts of the GIT and can be affected by different allogeneic and autogenous factors. Therefore, the microbiota intercedes in the digestion and absorption of nutrients, in cellular homeostasis, as a defensive barrier, with antioxidant, antimicrobial and immune properties and producing enzymes, vitamins and other nutrients that are deficient in diets [4].

Likewise, several studies have recommended intervening in the intestinal microbiota at an early age as a strategy to improve the productivity and animal health, especially considering the current production systems on farms [5,6]. These production systems (intensive) generate high risks of disease outbreaks due to pathogenic microorganisms, which promotes the use of subtherapeutic antibiotics, supplied on farms with dual purposes: to reduce mortality from enteric diseases and act as growth-promoters [7]. Antibiotics inhibit proteins with a broad spectrum for Gram-positive bacteria, in turn interfering with cell wall synthesis and peptidoglycan. On the other hand, ionospheres (type of soluble antibiotic) alter the concentration gradients in the cell membrane of Na^+^, H^+^ and K^+^ ions. However, the action of antibiotics on large Gram-positive bacteria is slower than for those that are Gram-negative [8,9].

Past indiscriminate use of antibiotics brought about resistance to many bacterial strains, a process that was enhanced by the ability of bacteria to transfer resistance, even between different genera and species [10]. Subtherapeutic antibiotics, beyond controlling pathogenic microorganisms, also affect many beneficial microorganisms, causing disturbances to the balance of the gastrointestinal microbiota. Many of these antibiotics or their residues can remain in animal tissues destined for human consumption [8]. Likewise, Iramiot et al. [11] mentioned that there is a high probability of multidrug-resistance transmission between humans and animals and that the carriage of multidrug-resistant bacteria between humans is 93%, followed by 80% between cattle, which represents a public health problem around the world.

Unquestionably, one of the alternatives to subtherapeutic antibiotics is probiotics; these natural products have shown efficacy, repeatability and safety in animals [12]. Probiotics are nutritional supplements made up of live microorganisms that colonize and modify the GIT microbiota [13], and in adequate amounts [14], they confer a benefit for the health and physiology of the host [15]. Many benefits of the use of probiotics have been described, such as protection against physiological stress, modulation of the intestinal microbiota, improvement of the epithelial barrier in the intestine and stimulation of the antioxidant capacity and immune system [16,17]. However, as disadvantages of the application of probiotics in animals, low repeatability of the benefits of some probiotic strains in different individuals, animal species and productive categories has been found, and in addition, some strains have a low tolerance to the temperature of feed manufacturing and chlorine in drinking water. Beyond this, the benefits of some probiotics are mediated by the type of substrate they receive in the gastrointestinal tract of the host. Hence, more conclusive studies on the benefits of probiotics in animals are still needed [17].

There are a great variety of microorganisms with probiotic characteristics—mainly these are lactic acid bacteria (LAB)—which actively participate in fermentation processes and several metabolic activities [18]. Those most often used are *Lactobacillus acidophilus*, *L. casei*, *Pediococcus pentosaceus*, *L. helveticus*, *L. lactis*, *L. salivarius*, *L. plantarum*, *Enterococcus faecium* and *E. faecalis* [17,19].

Furthermore, *Bacillus* spp.—sporulated Gram-positive bacteria of the Firmicutes division that do not colonize the GIT—are frequently used in animal production as probiotics [20]. The strains most often used are *B. subtilis*, *B. cereus*, *B. licheniformis*, *B. coagulans*, *B. polyfermenticus*, *B. pumilus* and *B. clausii*. These probiotics produce enzymes and vitamins and have antioxidant and microbial properties [21]. Live yeasts of the genus *Saccharomyces* have also shown probiotic activity in farm animals, modifying the intestinal microbiota, reducing the risk of dysbiosis and producing vitamins and enzymes [22,23]. Those most often used are *Saccharomyces boulardii* and *Saccharomyces cerevisiae* [24].

The microbial succession process shows that microorganisms can almost completely colonize the GIT weeks after birth [25,26,27]. Generally, microorganisms with probiotic characteristics are isolated from the large intestine by invasive methods and from feces by non-invasive methods in ruminants, poultry and pigs [28,29,30]. The FAO/WHO [15] emphasizes the importance of the specificity of the action and not of the source of origin of the strain. Some studies have recommended that to improve the efficacy of probiotics we can isolate native strains by region and even the animal species in which the product will be implemented, as well as use strain mixtures, genetic manipulations and synergistic components such as prebiotics. This can ensure competitive exclusion in the GIT and interaction between resident microorganisms [31,32].

However, Dowarah et al. [13] mentioned that not all strains of microbial species are effective for use as probiotics, meaning success will depend on the in vitro results, which predict their response in vivo in different experimental environments. The beneficial effects of probiotics are known to be specific to each microbial strain; therefore, it is important to identify probiotic strains and obtain reliable in vitro and in vivo results to extend the remit of these natural products to animal production [33]. Thus, it is necessary to observe the growth of probiotic candidate bacterial strains under changes in pH, salinity, NaCl, temperature and bile salts, as well as assess their antagonism with pathogenic bacteria and sensitivity to the use of antibiotics [34,35]. Likewise, in vitro results do not guarantee the viability and efficacy of probiotics in vivo as the hostile environments of the GIT may decrease the efficacy of probiotic strains. Therefore, it is necessary to carry out in vivo studies with apparently healthy and disease-challenged animals and under different stress conditions to evaluate whether beneficial microorganisms can colonize the gastrointestinal tract and/or modify the intestinal microbiota, and in turn, influence the biological indicators of animals [28].

The aim of compiling this review was to provide an updated summary of the probiotics’ role and their symbiotic relationship with the host animal, with emphasis on in vitro and in vivo studies in ruminants, pigs and poultry.

## 2. Methodology

### Search History

To conduct this review, which deals with probiotics and their response in animals of zootechnical interest such as ruminants, pigs and poultry, an electronic search was carried out in 2020–2021 of scientific articles from the last 10 years published in academic journals indexed in ISI, Web of Science (WoS) and Scopus.

The strategy for the search for scientific information was divided into four topics of interest: probiotics in in vitro studies; probiotics’ effect on the intestinal environment, probiotics’ effect on the antioxidant capacity and probiotics’ effect on the productivity and health of ruminants, pigs and poultry. The following eligibility criteria were considered: design of the study, relevance, probiotic candidate, dose, culture medium and variables determined in vitro and in vivo. Studies were excluded if they: were inconclusive, had low repeatability, lacked a hypothesis or compliance with it, presented incomplete materials and methods, gave little information on the strain used or lacked peer review.

Systematic searches were carried out of electronic databases of recognized academic prestige as such PubMed (https://pubmed.ncbi.nlm.nih.gov/) (accessed on 10 November 2021), Google Scholar (https://scholar.google.com) (accessed on 15 November 2021), ScienceDirect (https://www.sciencedirect.com/) (accessed on 17 November 2021) and NCBI-PCM (https://www.ncbi.nlm.nih.gov/pmc/) (accessed on 20 November 2021). The keywords used for the search were: antibiotic growth promoters, antimicrobial resistance and alternatives to subtherapeutic antibiotics; in vitro effect of probiotics on the inhibition of pathogenic bacteria, production of fatty acid chains, autoinduction (Quorum), bacteriocin production, pH, adherence, agglutination, methane production and immunomodulatory activity, as well as the growth of probiotics in different conditions of pH, bile salts, temperature, NaCl and antagonism; the role of lactic acid bacteria and yeasts in antioxidant activity, gut health, gut microbiota, immune response, productivity, meat quality, egg quality and health in ruminants, pigs and poultry.

This review article has 195 references: 192 articles published in peer-reviewed scientific journals, two chapters of scientific books from prestigious publishers and a reference from an international organization. For the effect of probiotics in in vitro studies, 20 references were used [30,36,37,38,39,40,41,42,43,44,45,46,47,48,49,50,51,52,53,54], which demonstrate the desirable probiotic characteristics of bacterial isolates for survival in the gastrointestinal tract, the production of metabolites such as bacteriocin and its antagonistic effect on pathogens. It is necessary to replicate the in vitro and in vivo results, as well as to find the specific mode of action (biochemical reactions) of each proposed probiotic candidate in the animal host. Likewise, 27 references [55,56,57,58,59,60,61,62,63,64,65,66,67,68,69,70,71,72,73,74,75,76,77,78,79,80,81] were used that support the importance of the microbiota and its symbiotic relationship with the host through physical and chemical processes. Likewise, to support this review of the role of probiotics in reducing free radicals and oxidative stress, 13 bibliographical references were used [63,82,83,84,85,86,87,88,89,90,91,92,93].

Likewise, the review covers how it is that some probiotic strains such as *Lactobacillus* spp., *Propionibacterium* spp., *Saccharomyces cerevisiae*, *Enterococcus* spp., *Bifidobacterium* spp. and *Bacillus* spp. have a direct influence on the characteristics of the microbiota, immune response, growth performance and meat quality. The references used to support this review of the biological function of probiotics in animals were: 27 references [94,95,96,97,98,99,100,101,102,103,104,105,106,107,108,109,110,111,112,113,114,115,116,117,118,119,120,121] that state the role of probiotic strains in ruminal function and in the productive response of ruminants. (Some studies are inconclusive even when using the same probiotic strains, especially for mastitis control. Thus, it is essential to continue this line of research to clearly define the doses of the probiotic and its effects on different production stages in ruminants.) In addition, 30 bibliographic references [27,29,79,122,123,124,125,126,127,128,129,130,131,132,133,134,135,136,137,138,139,140,141,142,143,144,145,146,147] were used to describe the oral administration of probiotics and their effect on growth performance, diarrheal syndrome and the meat quality of pigs. It is important to point out that more studies are still needed to corroborate the dose, optimal moment (before or after weaning) and the exposure time of the administration of probiotics in pigs, as well as the influence of probiotics on the organoleptic characteristics of pork. Our review of the effect of oral administration with probiotics on competitive exclusion, stimulation of the immune system and improvement of metabolic processes, such as the production of digestive enzymes and the absorption of nutrients in poultry, was supported by 49 scientific articles [8,49,148,149,150,151,152,153,154,155,156,157,158,159,160,161,162,163,164,165,166,167,168,169,170,171,172,173,174,175,176,177,178,179,180,181,182,183,184,185,186,187,188,189,190,191,192,193,194]. Despite various investigations of the use of probiotics in poultry, more research is necessary to establish the viability of these microorganisms in drinking water and feed, as well as to elucidate the biological benefits of a blend of strains or a single strain, considering the age, health condition, production conditions, genetic line and productive purpose.

## 3. Effect of Probiotics on In Vitro Results

Bacteria with probiotic characteristics have demonstrated bactericidal properties by producing antimicrobial substances such as organic acids, bacteriocin, lactoferrin and hydrogen peroxide [36], which is why probiotics are used to inhibit pathogenic bacteria, based on the results for in vitro methods [37]. Some probiotic bacteria (mainly LAB) can ferment disaccharides such as lactose and sucrose. That favors short-chain fatty acids’ production, such as acetic, propionic, butyric and lactic acids, which in turn, via photon emission, reduces the intestinal pH, thus promoting competitive exclusion and a significant reduction in the proliferation of pathogenic bacteria that do not survive at a relatively low pH [38,39].

We know that bacteria communicate intercellularly through chemical signals’ secretion. The so-called autoinductors are exacerbated according to the cell density, and this biochemical process called quorum affects bacterial cells and the host’s behavior. Quorum has been defined as the communication between bacteria of the same species (intraspecies communication), which reflects the ability of bacteria to monitor their population density through chemical signals and regulate gene expression [40,41]. Such is the case for probiotics, which can affect the proliferation of pathogenic bacteria through quorum sensing and influence their pathogenicity [42]. The formation of biofilms by probiotic strains in the GIT contributes to bacterial resistance and improves the exchange of nutrients among the microbiota and host, also reducing the adherence of pathogenic bacteria in the intestinal lumen [43]. In this sense, Kiymaci et al. [44] found that probiotic strains of *Pediococcus acidilactici* M7 inhibited the connectivity of molecules for quorum detection, as well as the virulence of the pathogenic strain *Pseudomonas aeruginosa*. Jiang et al. [45] also demonstrated that *Lactobacillus plantarum* intervened in the bacterial communication system (Quorum), perhaps due to the in vitro results of bile tolerance, antimicrobial effect and bacterial colonization in the GIT.

Betancur et al. [30] reported that *Lactobacillus plantarum* CAM6 isolated from the rectum of a Colombian Creole pig inhibits the in vitro growth of *Enterobacteriaceae* (*Klebsiella pneumoniae* ATCC BAA-1705D-5, *Pseudomonas aeruginosa* ATCC 15442, *S. enterica* serovar *Typhimurium* 4.5.12, *E. coli* strain NBRC 102203) and is resistant to the presence of growth-promoting antibiotics (ciprofloxacin, trimethoprim, tetracycline and doxycycline). Furthermore, this probiotic strain grew more than 10^6^ CFU/mL in response to an acidic or alkaline pH and at different temperatures, concentrations of bile salt and NaCl. Moreover, García et al. [46] reported that the *Lactobacillus pentosus* strain LB-31 demonstrated antagonistic activity and antimicrobial susceptibility, as well as tolerance to bile salts, pH changes and hydrophobicity. Furthermore, Abdelbagi et al. [47] reported that probiotic strains (encapsulated or not) in in vitro studies decreased methane production and increased the total gas production in relation to the control treatment.

On the other hand, the mode of action for probiotic yeasts is different from that for bacteria. Thus, in vitro tests must be performed such as those assessing the ability to adhere and co-aggregate with pathogenic microorganisms, which simulates the possible expulsion of pathogenic microorganisms from the GIT [48]. In this sense, Rodriguez et al. [49] reported that an enzymatic hydrolysate from *Saccharomyces cerevisiae* decreased the in vitro growth of *E. coli*, *Staphylococcus* spp., *Salmonella* spp. and *Klebsiella* spp. by producing bacteriocins. Furthermore, this natural product induced the co-aggregation (adhesion of bacteria to each other or different types, forming a biofilm on already established bacteria and facilitating the adhesion of secondary colonizers) of these pathogenic bacteria to the components of its cell wall [50]. It is important to note that live yeast increases the beneficial population growth in the gut by stimulating the immune response [51]; however, studies indicate that some freeze-dried (dead) yeasts decrease the size of their population in the fermentation process. Unlike live yeasts, the mode of action of dead yeasts is based on bioactive compounds (β-glucan, chitin and nucleic acids) secreted or isolated from their cell wall with immunostimulatory properties, which contribute to the morphology of microvilli and cell differentiation [51,52].

García et al. [53] isolated probiotic yeasts from broiler feces such as *Kodamaea ohmeri*, *Trichosporon asahii*, *Trichosporon* spp., *Pichia kudriavzevii* and *Wickerhamomyces anomalus*. These strains demonstrated resistance to low pHs and bile salt concentrations, and *W. anomalus* had the highest agglutination and adherence capacity, meaning it reduced the pH and demonstrated an ability to grow under stress conditions. In a similar study, Fernandes et al. [52] isolated yeasts with probiotic potential such as *Magnusiomyces capitatus*, *Candida ethanolica*, *Candida paraugosa*, *Candida rugosa* and *Pichia kudriavzevii* from ruminal liquid and obtained positive results in reducing the pH, acid accumulation and neutral detergent fiber digestibility, which was important for simulating the survival of these microorganisms in the ruminal environment. Moreover, a new probiotic formulation based on the yeast *Debaryomyces hansenii* showed strong immunomodulatory activity in in vitro studies due to polyamines and wall-like component β-D-glucan [51]. This modulates the composition of the microbiota and inhibits the proliferation of harmful bacteria, which improves the phagocytic activity of macrophages and the antimicrobial activity of mononuclear cells and neutrophils [54]. The effects of probiotic yeast in in vitro studies are summarized in Table 1.

## 4. Effect of Probiotics on Intestinal Environment

The layer of cells that make up the epithelial tissue that covers the GIT represents the greatest relationship between the interior and exterior of the host [55]. Additionally, these cells contribute to the symbiotic relationship between host and microbiota through immunological secretions, bacterial antigens and both physical and chemical mucosal barriers [56]. The balance of the intestinal environment is rapidly altered due to stress, medicaments, infections and diet changes, among others [57].

The GI tissue maintains several defense pathways, one of which is via the binding proteins, made up of many proteins found in the permeable and selective barrier of the cell membrane. Liu et al. [58] found that a higher genetic expression of zonula-occludens-1 and occludin favored the absorption capacity and reduced diarrheal syndrome in pigs. We know the relationship between the intestinal barrier and the binding proteins: a decrease in the expression of these proteins causes an uncontrolled entry of macromolecules, undesirable nutrients and microorganisms in the intestinal lumen, which reduces the absorption of nutrients and the animal response [59]. Thus, probiotics and their metabolites can regulate the expression of binding proteins and improve the selective capacity of the intestinal barrier, along with the intestinal health of the host, by inhibiting the effect of lipopolysaccharides that induce an inflammatory response of cytokines, thus preventing the destruction of zonula-occludens [60,61]. In this sense, Liu et al. [61] demonstrated that a probiotic strain *Enterococcus faecium* HDRsEf1 increased the genetic expression of binding proteins (especially zonula-occludens-1) by inhibiting the pro-inflammatory response mediated by decreased TNF-α production. Moreover, strains of *Bacillus* spp. (*subtilis* and *pumilus*) on broiler diets raised under stressful conditions increased the expression of occludin, zonula-occludens-1 and junctional molecule binding [62]. Furthermore, Izuddin et al. [63] reported that oral administration of probiotics in lambs increased binding proteins such as occludin, claudin-1 and CLDN4, which favor the functioning of the ruminal barrier. Likewise, studies have shown that probiotics can stimulate the immune system of the host mucosa through interaction between receptor recognition patterns signaling from GIT cells and molecular patterns associated with microbes from probiotic strains [64]. According to Yousefi et al. [65], some probiotic strains have immune activity because they can modulate cytokine production and increase mucin secretion, phagocytosis, natural killer T cells’ activity and IgA production.

The morphology of the villi and crypts is also directly related to the intestinal function and morpho-physiology of animals [66]. A diet based on probiotics possibly improves the development of the intestinal epithelial tissue, however, the cilia height in the intestine can change in size depending on the species of microorganism that interact [67]. Probiotics can increase the villi height (VH) and the crypt depth (CD), and in turn, improve the productive efficiency and absorption capacity of nutrients and reduce the metabolic requirements [68]. Several studies with probiotic strains such as *L. acidophilus* [69], *L. plantarum* 22F and 25F [70] and *Pediococcus acidilactici* FT28 [71] related the non-antibiotic growth-promoting effect of these nutraceutical products to the increase in villi height, crypt depth and the ratio of VH to CD in the small intestine of animals.

The use of probiotics can provide cellular defense mechanisms by inducing anti-inflammatory cytokines [72]. Furthermore, lipoteichoic acid, as a component of the cell wall associated with the surface of Gram-positive bacteria such as bifidobacteria and lactobacilli, stimulates NO synthetase and pathogen-infected cell-death mechanisms [73]. Some *Lactobacillus* strains can incorporate cytokines to act as immunological modulators by enhancing the activity of macrophages, antibodies and apoptotic cell activation [74]. Likewise, Galdeano et al. [75] found that *Lactobacillus sakei* and *Lactobacillus johnsonii* changed the expression of IL-1β, IL-8 and TNF-α and TGF-β, respectively, which stimulated the production of IgA, IgM and IgG.

The antidiarrheal effect of some probiotic strains has justified their commercial use in mammals [76]. Diarrhea is known to be caused mainly by Enterobacteriaceae such as *Salmonella* spp. and *Escherichia coli* [77] and exacerbated by stressful situations. A study by Lee et al. [78] found that probiotic strains of *L. plantarum* decreased the diarrhea degree in piglets challenged with enterotoxigenic *Escherichia coli* (ETEC) due to competitive exclusion in the GIT. Betancur et al. [79] also reported that oral administration of probiotic strains reduced the incidence of semi-liquid, liquid and pasty feces in piglets, mainly due to a higher production of volatile fatty acids and a decrease in intestinal pH.

Furthermore, other studies showed that the use of multispecies probiotics (*Pediococcus acidilactici*, *Enterococcus faecium*, *Lactobacillus acidophilus*, *Lactobacillus casei* and *Bifidobacterium bifidum*) decreased the incidence and duration of metabolic diarrhea in young calves, which has direct implications on gut health. However, these feed products did not affect the performance of the animals [80]. Likewise, Wu et al. [81] confirmed that a mixture of multispecies probiotics, at a ratio of 2 g/day/animal during the first four months of life, decreased the relative abundance of Bacteroidetes and increased the relative abundance mainly of *Bifidobacterium* and *Lactobacillus*, which reduced the diarrhea incidence and promoted the weight gain of calves.

## 5. Effect of Probiotics on Antioxidant Capacity

In the modern agricultural system, which is intensive, animals are frequently exposed to oxidative stress. It is known that the imbalance between oxidants and antioxidants due to the overproduction of reactive oxygen species (ROS) and nitrogen (RNS) provokes oxidative stress [82,83]. The physiological concentration of ROS is essential to maintain normal cellular functions [84]. An uncontrolled increase can cause cellular damage [85], but physiologically, the negative effects can generally be neutralized by endogenous antioxidants such as superoxide dismutase, catalase, ascorbic acid and heat-shock proteins [86]. Added to this, Izuddin et al. [63] demonstrated in an in vivo study that some probiotic strains could reduce the excessive concentration of reactive radicals, which contributes to reducing the risks of various diseases associated with lipid peroxidation and oxidative stress.

Thus, one of the strategies to reduce ROS activity and oxidative stress is oral administration with lactobacilli that convert superoxide radicals into oxygen and hydrogen peroxide [87,88]. Moreover, many of the LAB species produce catalase that scavenges hydrogen peroxide, which blocks the formation of peroxyl radicals [89]. Likewise, other bacterial strains (mainly lactobacilli) decrease reactive oxygen intermediates through the production of glutathione and thioredoxin, considered endogenous antioxidants [90]. Sosa-Coccio et al. [34] found that *Lactobacillus pentosus* LB-31 had an antioxidant response in vivo by increasing the serum glutathione concentration by 23.85% in broilers.

In this way, Dowarah et al. [91], using *Pediococcus acidilactici* FT28 and *Lactobacillus acidophilus* NCDC15 as probiotic candidates, found increased serum concentrations of catalase, superoxide dismutase and glutathione peroxidase in weaned pigs. Likewise, Xiang et al. [92], when testing several probiotic strains such as *Clostridium butyricum* and *Saccharomyces boulardii,* plus *Pediococcus acidilactici* in laying hens, reported a decrease in the concentration of malondialdehyde in the serum and reactive oxygen species (ROS) in the ileum and cecum with the first bacterial strain (*C. butyricum*). Furthermore, Yang et al. [88] demonstrated that the genus *Lactobacillus* as *Bifidobacterium longum* and *L. fermentum* can produce antioxidant compounds in vitro and in vivo, which could counteract oxidative stress in the host. Moreover, Hou et al. [87] showed that *L. reuteri* strains cause positive changes to the concentration of superoxide dismutase and glutathione peroxidase, which was the scientific justification for their growth-promoting effect in pigs. Likewise, Amaretti et al. [93] reported that probiotic strains such as *Lactobacillus brevis* DSMZ 23034, *Lactobacillus acidophilus* DSMZ 23033 and *Bifidobacterium animalis* subsp. *lactis* DSMZ 23032 have high antioxidant potential in an animal model challenged with doxorubicin. The authors concluded that the colonization of probiotic bacteria promoted intestinal saccharolytic metabolism and increased the concentration of endogenous antioxidant enzymes.

## 6. Effect of Probiotics on In Vivo Studies

### 6.1. Ruminants

Ruminants consume a great variety of substrates in their diet, which are not digestible by other mammals and poultry. The complex fermentation process mediated by the ruminal microbiota (10^8^ to 10^11^ CFU/mL of ruminal content) produces energy, proteins and vitamins (water-soluble vitamins and vitamin K) of microbial origin, which are essential for milk production and/or weight gain [94]. A method to manipulate the microbiota of the rumen during its growth period is to directly provide activators and/or probiotics, to establish a balance in the microbiota, which is more efficient during growth than in adults [95]. LAB and yeasts (*S. cerevisiae*) are used as ruminal activators/probiotics for their ability to affect the dynamics of the microbiota in the rumen and the way in which nutrients are decomposed [96,97]. In this sense, Jinturkar et al. [98] confirmed that individual or combined supplementation with *Saccharomyces cerevisiae* and *Lactobacillus acidophilus* improved the growth performance of growing goats.

Also, Abd El-Trwab et al. [99] demonstrated how probiotics can improve the ruminal function due to the proliferation of ruminal microorganisms, which provokes a tolerable reduction in ruminal pH and a higher concentration of volatile fatty acids. Maldonado et al. [100] also summarized how the manipulation of the interaction between the intestinal microbiota and the host (ruminants) results in improvements to the ruminal activity, nutrient digestibility and animal response. Likewise, LABs and yeasts are used to increase the population of cellulolytic bacteria (capable of producing cellulase) in the rumen [101,102,103], which significantly affects the degradation of cellulose, thus improving microbial fermentation, and therefore, protein synthesis [104].

In this sense, the use of *Lactobacillus casei* and *Lactobacillus plantarum* P-8 improves milk production and the concentration of lysozyme, lactoperoxidase and lactoferrin, with a notable decrease in somatic cells, although without changes to the chemical composition of the milk and without modifying the bacterial diversity of the rumen [105]. Additionally, studies by Stein et al. [106] and Stella et al. [107] demonstrated how including *Propionibacterium* spp. and *Saccharomyces cerevisiae* in dairy cows and dairy goats improved the feed conversion rate, milk production and dry matter intake, respectively. Cruywagen et al. [108] also recommended using 1 mL (5 × 10^7^ CFU) with *Lactobacillus acidophilus* as a milk substitute since it does not reduce the body weight, dry matter intake or productive efficiency of Holstein-Friesian calves at two weeks old. Furthermore, Apás et al. [109] reported that a compound of several probiotic strains such as *Lactobacillus reuteri* DDL 19, *Lactobacillus alimentarius* DDL 48, *Enterococcus faecium* DDE 39 and *Bifidobacterium bifidum* modified the microbial environment, and in turn, the acid profile of goat milk, with an increase in unsaturated fatty acids, mainly linoleic, linolenic and conjugated linoleic, in addition to a decrease in the atherogenic index. Further to this, Ma et al. [110], using *Enterococcus faecalis, Bacillus subtilis* and *Saccharomyces cerevisiae* as probiotics, found positive responses in terms of the feed intake, milk production and intestinal microbiota (mainly *Succinivibrionaceae)* of Saanen dairy goats.

Ruminal activators/probiotics are effective at improving the health of ruminants [99]. In this sense, probiotic strains used as therapeutics can reduce or almost eliminate the proliferation of *E. coli* 0111: NM8 and *E. coli* 0157: H7 in the GIT of weaned calves, which reduces the incidence of enterohemorrhagic *E. coli* [111]. Likewise, Stanford et al. [112] reported that a microbial activator with *Lactobacillus casei*, *Lactobacillus lactis* and *Paenibacillus polymyxa* decreased the spread, frequency and prevalence of *Escherichia coli* O157: H7 in the stool of cattle. The authors recommended using these beneficial microorganisms up to two weeks before slaughter to reduce the risk of carcass contamination. Dysbiosis of the commensal intramammary microbiota and the pathogenic bacteria that cause mastitis also influence the presence of breast inflammation [113]. Though poorly documented, it appears that lactic acid bacteria that produce bacteriocin or have the ability to adhere to cells can change the microbiota of the teat apex and/or decrease the proliferation of pathogenic bacteria present in it, and in turn, prevent or a gradually treat mastitis, leading some authors to suggest the direct use of bacteriocin in the teat apex as a treatment for mastitis [114,115]. In this sense, isolates of the genera *Lactobacillus* and *Lactococcus* have been evaluated based on topical application at the teat apex, showing the inhibitory activities of *S. aureus*, *S. uberis* and *E. coli* by colonizing the epithelium and competing with pathogenic bacteria [115].

Other studies did not recommend the use of probiotics as they found that beneficial bacterial strains induced inflammation of the mammary glands, with a high concentration of neutrophils in the milk. It seems the effects of probiotics on mastitis depend on the degree of mastitis, number of somatic cells, degree of inflammation in mammary glands and type of bacterial strain.

To meet the high nutritional needs of cows during lactation, diets are often low in physically effective dietary fiber and rich in concentrates, leading to rumen acidosis [116]. Due to the changes produced by probiotics in the ruminal microbiota, these have been used as natural alternatives to prevent or treat predisposing causes of acidosis [117]. In this spirit, Lettat et al. [118] administered *Propionibacterium* P63, *L. plantarum* and *L. rhamnosus* strains through an intraruminal cannula at a rate of 1 × 10^11^ CFU/animal/day, and managed to stabilize the pH of the rumen and prevent acidosis in sheep. Han et al. [117] also found that oral administration of three probiotic yeasts increased the variety of ruminal microflora and decreased metabolic acidosis and inflammation in sheep. Likewise, oral induction with a multi-strain probiotic reduced the ruminal pH and concentration of lactic acid in the ruminal fluid, thus preventing acidosis [119]. Furthermore, Han et al. [117] found that supplementation with yeast *S. cerevisiae* stabilized the ruminal pH and acid-base balance, which increased the diversity of ruminal microflora, with a decrease in intestinal inflammation and acidosis in sheep. However, other studies with *S. cerevisiae* cultures did not find a positive response in ruminal fermentation, nor the possibility of preventing ruminal acidosis [120]. Some studies suggest that the administration of probiotics helps the ruminal microbiota to adapt to the presence of lactic acid, in addition to increasing the population of lactic acid bacteria, which is related to a better capacity for carbohydrate fermentation and absorption of volatile fatty acids, resulting in an increase in ruminal pH [121].

It seems the positive effects of probiotics on ruminal acidosis depend on the strain and concentration of the probiotic, along with the age, diet, acid-base balance, health status and production technology. The benefits of probiotics in ruminants are summarized in Table 2.

### 6.2. Pigs

We know that microbial succession occurs via the colonization of the sterile intestine with microorganisms of vertical and horizontal origin in the weeks after birth [27]. Due to the intensive production model, weaning in piglets can be a critical period due to changes in the diet, separation from the mother and the environment [122]. This process is associated with the development of infections; pathogenic microorganisms can induce microbial dysbiosis by entering and colonizing the GIT, causing the formation of severe ulcers, constipation, intestinal inflammation and diarrhea, which causes poor absorption of nutrients, poor feed conversion and high mortality [123].

Pig feeding represents approximately two-thirds of total production. Therefore, increasing the feed efficiency is vital for production profitability [124]. Since it is necessary to reduce high costs in production, managing the intestinal microbiota is a potential strategy to avoid health problems, reduce diarrhea and increase the yield, especially in modern production systems [125]. Hence, in pig production, supplemented antibiotics in feed and drinking water have been used to modulate the intestinal microbiota in different stress situations. However, these provoke a residual effect in animals, trigger antibiotic resistance in bacteria and provoke the risk of disease in humans [126]. As an alternative to subtherapeutic antibiotics, probiotics are one of the most viable alternatives in pigs [127,128]. Thus, probiotics are used in all stages of pig production (with an emphasis on young pigs) such as reproduction, transition and fattening [129].

Some types of probiotics produce dietary enzymes such as lipase, amylase, protease, cellulase and phytase, which are important for the absorption of nutrients, including some that are not digestible by non-ruminant animals; in addition, they promote growth performance, nutritional efficiency, intestinal health, capacity antioxidant and the immune system in pigs [130,131,132]. Further to this, probiotics added to the diet produce beneficial fermentation that provokes an increase in the concentrations of SCFAs and lactic acid in the GIT, which lowers the intestinal pH and hinders the growth of opportunistic enteric pathogens that need a slightly acidic or alkaline medium to grow and multiply [133].

A study in pigs showed that a single oral dose at 5 × 10^10^ CFU/mL or 5 × 10^9^ CFU/mL in suckling piglets or at weaning with *L. plantarum* (DSMV 8862) or *L. plantarum* (DSMZ 8866), respectively, changed the microbial population with competitive exclusion in the small and large intestines, which improved the productive response [134]. It seems the oral administration of probiotics in piglets weaning for between 25 and 28 days could reduce post-weaning stress, which is provoked by the sudden change from a liquid diet to a solid diet and the increase in the population of pathogenic bacteria opportunists (*Enterobacteriaceae*) [133,134]. Furthermore, Betancur et al. [135] found that oral administration with *L. plantarum* CAM6 (5 × 10^6^), compared to a dietary antibiotic, promoted a better body weight and feed efficiency, and also increased the serum IgA concentration, although no significant changes were noted in the hemogram of the weaned piglets.

Microbial genera such as *Lactobacillus*, *Bacillus* and *Streptococcus* benefit the colostrum quality, milk quality and milk yield [136]. In addition, probiotic action directly influences the number of weaned piglets and the live weight at weaning, as well as decreasing diarrhea in the lactation process [137]. Alexopoulos et al. [138] showed that *Bacillus licheniformis* and *Bacillus subtilis* spores improved the composition and quality of milk in weaning sows. In turn, the use of 10^10^ CFU of *Lactobacillus fermentum* LFQI6 supplied from day 80 of gestation decreased the weight loss of the breeders and the mortality of the piglets, as well as significantly improving the body weight of the litter [139]. Furthermore, Betancur et al. [79] demonstrated how oral inclusion of *Lactobacillus plantarum* CAM6 in sows improved the lactose concentration in milk and decreased diarrhea and piglet mortality, which promoted a better body weight and increased the Na+, D-β-hydroxybutyrate, leukocytes and lymphocytes in the blood. In addition, Ayala et al. [140], in a study with *Bacillus subtilis* and its endospores, found an increase in milk production and plasma proteins, as well as lower weight loss after lactation, in breeding Yorkshire-Landrace × L35 sows.

Swine production systems are intended to favor the carcass yield with fast-growing pigs by emphasizing aspects that indicate the state of health and metabolism of the animal [141]. Yet, modern pig production systems are frequently exposed to oxidative stress and damage that can affect the meat quality [29]. Fattening pigs maintain a higher immunity than piglets, which is why they resist problems related to intestinal pathologies. Even so, we know that using probiotics can enhance the growth and improve the performance and meat quality [142]. Tufarelli et al. [143] reported that the oral inclusion (100 mg/kg of body weight) of a probiotic blend (*Streptococcus thermophilus* DSM 32245, mixture of two strains *Bifidobacterium animalis* ssp. lactis DSM 32246 and DSM 32247, *Lactobacillus acidophilus* DSM 32241, *Lactobacillus helveticus* DSM 32242, *Lactobacillus paracasei* DSM 32243, *Lactobacillus plantarum* DSM 32244 and *Lactobacillus brevis* DSM 27961) in fattening pigs increased the concentrations of polyunsaturated fatty acids (ΣPUFA) and protein in pig meat, perhaps due to better gut health, digestibility and nutrient translocation, while other meat indicators did not change for the experimental groups. A probiotic strain based on *Pediococcus acidilactici* FT28 directly influenced the meat quality by improving the 2-thiobarbituric acid reactive substances, water retention capacity and pH of pork [91].

Likewise, Meng et al. [144] found that probiotic groups led to higher meat colorimetry, mainly in terms of the redness values and marbling scores. Černauskienė et al. [145] also demonstrated how consuming probiotics (*Enterococcus faecium*) improved the meat quality and organoleptic properties such as the colorimetry, fat infiltration and the firmness of the meat. Likewise, Suo et al. [146], using *L. plantarum* (1 × 10^9^ CFU/day), a probiotic isolated from the feces of weaned piglets, found an improvement in the meat pH and in various indices of meat texture such as chewiness, restoring force, hardness, stickiness and gumminess. The authors justified these results as stemming from competitive exclusion due to colonization of the probiotic strain and an increase in the villi height, which favor the absorption of nutrients. Furthermore, Chang et al. [147] revealed how oral administration of *Lactobacillus plantarum* (2.2 × 10^8^ CFU/mL) in the *Longissimus dorsi* muscle increased the concentrations of ascorbic acid, thiamine, amino acids such as serine, lysine, histidine and arginine, monounsaturated and polyunsaturated fatty acids, linolenic acid, linoleic acid and muscle yellowness (b*). Likewise, the probiotic group reduced the redness (a*), shear force, ashes and pH of pork meat. Currently, certain strains of probiotics are marketed with a focus on how they improve the intestinal physiology, immune system, intestinal health, productivity and swine reproduction. Table 3 indicates the functional properties of probiotics in pig production.

### 6.3. Poultry

The GIT of poultry has specific characteristics, such as crop, proventriculus, gizzards, two cecum and cloaca (reproductive, excretory and productive functions). Moreover, the gastrointestinal transit is shorter than that of mammals and is highly colonized by microorganisms that interact directly with the host [148,149]. Fermentation takes place in the ceca of poultry [150] because a complex microbiome is housed in this part of the GIT, which is densely populated with beneficial bacteria [151]. Hence, manipulating the microbiota by orally administering nutrients/supplements directly affects the intestinal morphology, nutrient digestibility and metabolic processes [152]. In the case of poultry, their GIT is not sterile; it is immature and must undergo important changes to its morphology and biochemistry, the most important being in the first 24 h of life [153]. This is since at the time of birth, the microbiota colonize it via various routes such as: transfer from the mother through the oviduct and through the pores in the eggshell [154]. Chickens are capable of inoculating microorganisms such as *Lactobacillus, Clostridium* and *Propionibacterium* prior to shell formation [155].

Poultry in intensive production, unlike birds in alternative production systems, do not have a sufficiently colonized intestinal tract [156]. Therefore, inoculation of the microbiota occurs after hatching and when exposed to the environment by going through incubation, transport, handling and vaccination processes, where pathogens can be harbored [157]. While wild animals have extensive production systems, their GIT is colonized by microorganisms from the environment, generating a beneficial symbiotic process [158]. The possibility of obtaining native microorganisms in intensive production systems is very low due to the high incidence of Enterobacteriaceae such as *E. coli* and *Salmonella* spp. These harmful microorganisms provoke infections and inflammatory responses, with significant economic losses [153,158]. A significant change occurs when poultry are exposed to a different environment in terms of the farm and diet, whereby the initial composition of the microbiota is mainly *Lactobacillus*. In the second week, the GIT matures, with a greater number of microorganisms in the intestine and cecum, due to the pH, anaerobiosis and bacterial metabolites such as short-chain fatty acids. Finally, at 21 days, there is an exponential growth of *Lactobacillus* spp. in the digestive tract, above 10^10^ CFU/g [159].

Growth-promoting antibiotics can reduce the intestinal microbiota by inhibiting the production of catabolic mediators that are activated by metabolic stress processes, which induce intestinal inflammation. On the contrary, the use of probiotics increases the beneficial microbiota by improving the intestinal barrier, which in turn, stimulates the immune system and competitive exclusion, thus improving the performance by absorbing nutrients properly [160,161]. Many studies have tested probiotics in broilers, laying hens, pullets and breeders, focused on improving the intestinal health, nutrient digestibility and productivity response and preventing bacterial infection in stressful situations. Probiotics have also been used to a lesser extent as a clinical treatment [162]. In this sense, Jin et al. [163], using *Lactobacillus acidophilus* and a mixture of 12 *Lactobacillus* spp., found improvements in the body weight and feed conversion ratio versus the control group. Yet, despite the improvement in yield, it is estimated that the use of multiple strains potentially increases the production costs beyond those lost due to mortality [8].

Furthermore, Afsharmanesh et al. [164] found that the use of 1 g/kg of a probiotic (8 × 10^5^ CFU of *Bacillus subtilis*/g) in broilers increased the height of the villi in the duodenum (1.40 vs. 1.51 mm), without changes to the morphometry of the jejunum and ileum. These results demonstrate that nutrient absorption and transport can be more efficient when probiotics are supplied, mediated by increased enzyme activity and a reduction of toxic chemicals [164,165]. Trials were carried out by adding probiotics to the diet (with *B. subtilis*, *L. salivarius*, *P. parvulus* [166] and *E. faecium* [167,168]), as a result, an increase in the villi height and proportion of villi/crypt depth in the ileum was observed in broilers, which resulted in a higher absorption capacity of the intestinal lumen, and in turn, a better productive performance of the host [169].

Birds are known to have the ability to mix urine with feces because the cloaca has three folds: the coprodeum, urodeum and proctodeum [170]. Denbow [171] demonstrated that probiotic strains can contribute to nitrogen metabolism by stimulating retrograde peristalsis in the host’s rectum. GIT bacteria can also represent an important source of non-essential amino acids. These beneficial microorganisms use biochemical reactions to produce amino acids from non-specific nitrogen stimulated by energy from endogenous fermentable carbohydrates [172]. According to Vispo and Karasov [173], *Lactobacillus casei* can convert uric acid into ammonia, which can be used by the host to synthesize some amino acids such as glutamine. Furthermore, it was reported that microbial diversity due to oral administration of *Bacillus* spp. influenced several metabolic pathways such as B6 metabolism, retinol metabolism and phosphonate metabolism [174].

Likewise, the colonization of certain bacteria such as *Bifidobacterium* and *Lactobacillus* in the GIT could increase the activity of the digestive enzyme; however, the proliferation of *Escherichia coli* can affect the secretion of these enzymes and damage the villi of the mucosa [175]. Manipulation of the gut microbiome by oral administration of probiotics may influence the antibody-mediated immune response. In this sense, probiotics containing *Streptococcus faecalis*, *Bifidobacterium bifidum* and *L. acidophilus* improved the systemic antibody response to red blood cells [176]. In addition, Khochamit et al. [177] reported that a mixture of probiotics with lactic acid bacteria increases the immune response due to increased genetic expression of IL-1β, IL-2 and IFNγ.

Some strains such as *Lactobacillus salivarius* NRRL B-30514 isolated from chicken ceca have been used to produce bacteriocin, and the results showed a reduction in the presence of *Campylobacter jejuni* in the intestine, while the use of *Enterococcus durans* managed to reduce the *Campylobacter* spp. to undetectable levels [178]. Furthermore, probiotics decrease the expression of IL-12 and IFN-γ and increase gut protection in broilers challenged with *Salmonella enterica* serovar typhimurium (TNF-α) [179]. The acidification of the intestinal environment by probiotics, achieved by producing compounds such as lactic and acetic acid, maintains an environment incapable of allowing the growth of enteropathogens [180].

Probiotics have also sparked interest in research on in ovo administration. Pender et al. [181] found no improvement in hatchability indicators when they used probiotics in ovo, although one week after hatching, productivity increased due to modulation of gene expression in the ileum. Likewise, in ovo injection with *Bacillus subtilis* stimulated microbial diversity in the GIT and increased amniotic fluid, and in turn, the genetic expression of MUC2 in the ileum, although without benefits for the productivity or immune response in broilers [182]. In ovo administration with probiotics in chickens challenged with coccidia decreased macroscopic lesions and mortality in broilers, which could represent a strategy to reduce the negative impact of this parasitic disease [183].

Although probiotics have been used less widely in laying hens than in broilers because the latter’s productive category is more susceptible, studies have shown that the use of probiotics (*Bacillus subtilis*) has a positive effect on production and egg quality for laying hens [184]. In this spirit, Macit et al. [185] found that probiotic strains increased the egg production, yolk color and monounsaturated fatty-acid profile. In a meta-analysis study, probiotics were reported to positively influence the shell thickness, shell weight and yolk color (*p* <0.01), as well as the total cholesterol, low-density lipoproteins and serum high-density lipoproteins [186]. Furthermore, in other economically important poultry species such as quail, the use of probiotics was shown to improve the enzyme activity, productivity, egg quality, fertility and hatchability [187].

On the other hand, one of the natural alternatives to the indiscriminate use of sub-therapeutic antibiotics in poultry production is yeasts [188]. There are approximately 2500 species of yeast in nature. Those most used either in a liquid or dry form in poultry nutrition are torula yeast and *Saccharomyces cerevisiae* [189]. Researchers consider that live yeasts play a probiotic role and dead yeasts have a prebiotic effect (mainly due to the cell wall) [190]. In poultry, oral administration of yeasts can reduce the proliferation of *Enterobacteriaceae* in the GIT, stimulate the immune system and improve metabolic processes, such as the production of digestive enzymes and absorption of nutrients [191]. In this sense, Feye et al. [192] found that yeast-fermentation products can reduce the virulence of some bacteria and improve antibiotic resistance, and this effect also can change the phenotypic characteristics of the pathogen. Moreover, in a study with *Saccharomyces cerevisiae* yeasts in young turkeys, the authors showed that the body weight increased from the first experimental week to the end of the experiment, as well as there being a decrease in the number of goblet cells, although without changes to the structure of the villi [193].

Likewise, interestingly, the yeasts could be used as part of the animal ration. In this spirit, Rodríguez et al. [49], using up to 20% dry torula vinasse yeast rich in protein (43.24%), amino acids, ash (7.15%) and B complex vitamins (mainly B_12_), with low percentages of crude fiber (1.20%) and with 2811 kcal/kg of metabolizable energy corrected for N, replaced 19.92% and 20% of soybean meal in the diets of broilers in the starter (0–21 days) and finisher (22–42 days) staged, respectively, without notable changes to the feed conversion ratio or yield of edible portions between experimental treatments. Similar results were reported by Rodríguez et al. [194] when they recommended the use of 20% dried torula yeast vinasse in layer pullets, which improved the feed efficiency relative to the control diet. The role of probiotics in poultry production is summarized in Table 4.

Studies have shown that probiotics will likely be part of preventive and therapeutic treatments applied as part of future health services. Therefore, it is necessary to focus our research on enhancing the ability of probiotics to modify the intestinal microbiota and improve the intestinal health and biological activity of the host. Currently, health challenges continue to increase, and probiotics could play a decisive role in increasing resistance to damage from pathogens that commonly inhabit the intestinal microflora and from other external microorganisms. It has further been suggested that probiotics could be used as part of the treatment against some emerging diseases in animals and against COVID-19 infection, which triggers an immune response in humans [195].

## 7. Conclusions

Probiotics unquestionably represent effective alternatives to the indiscriminate use of promoter antibiotics on animal farms. They have beneficial effects for health and production in the host animal by modulating the host microflora, inhibiting antimicrobial action, wielding antioxidant and immunological effects and affecting the intestinal morphology. Probiotic effects depend on the strain used, concentration, health state, diet, age, animal and productive category. Oral administration of probiotics improves the growth performance, feed conversion rate, nutrient utilization, gut microbiota and gut health and reduces diarrheal syndrome and bacterial infections in ruminants and non-ruminants.

## Figures and Tables

**Table 1 animals-12-00719-t001:** Effects of probiotic yeasts in in vitro studies.

Yeast Strain(s)	Type	Effect	Reference
*Saccharomyces cerevisiae*	Lyophilized	Reduces the growth of *E. coli*, *Staphylococcus* spp., *Salmonella* spp. and *Klebsiella* spp.	[49]
*Kodamaea ohmeri*, *Trichosporon asahii*, *Trichosporon* spp. *Pichia kudriavzevii* and *Wickerhamomyces anomalus*	Live yeast	Grows at low pHs and high concentrations of bile salts	[53]
		High adherence and agglutination capacity, reduces intestinal pH and grows under stress conditions	
*Magnusiomyces capitatus*, *Candida ethanolica*, *Candida paraugosa*, *Candida rugosa* and *Pichia kudriavzevii* *Saccharomyces cerevisiae*	Live yeast	Reduces intestinal pH and acid build-up and increases the digestibility of neutral detergent fiber	[52]
	Lyophilized commercial yeast	Rapidly reduces yeast population during the first 12 h of fermentation (growth test)	
*Debaryomyces hansenii*	Live yeast	High immunomodulatory activity	[51]

**Table 2 animals-12-00719-t002:** Effects of probiotics on ruminant production.

Strain(s)	Cell Count	Mode of Administration/Dose	Host/Duration	Effect	Reference
*Lactobacillus acidophillus* and *Saccharomyces cerevisiae*		Individually (2 g) and combination of both in the feed (1 g of each)	Goats (35 days)	Increases the average daily weight gain	[98]
*Lactobacillus casei* and *Lactobacillus plantarum*	1.3 × 10^9^ CFU/g	Combination of both in the feed (50 g/day)	Dairy cows (30 days)	Increases the milk production and the contents of milk immunoglobulin G, lactoferrin, lysozyme and lactoperoxidase	[105]
*Propionibacterium* spp. and *Saccharomyces cerevisiae*	6 × 10^11^ CFU/cow	Orally, mixed in feed	Dairy cows (25th week of lactation)	Improves the feed conversion rate, milk production and dry matter intake	[106]
*Saccharomyces cerevisiae*	4 × 10^9^ CFU/day	Orally, mixed in feed (0.2 g/day)	Dairy goats (15th week)	Improves the feed conversion rate, milk production and dry matter intake	[107]
*Lactobacillus acidophilus*	5 × 10^7^ CFU/mL at each of two feeds/day	Orally, mixed in feed	Holstein-Friesian calves	Regulates body weight under milk-replacer conditions	[108]
*Lactobacillus reuteri, Lactobacillus alimentarius, Enterococcus faecium* and *Bifidobacterium* *bifidum*	10^9^ CFU/mL resuspended in milk	Orally, resuspended in milk (1 mL/two feeds per day)	Goats (42 days)	Improves the microbial environment and intestinal health, as well as the acid profile of milk, with an increase in unsaturated fatty acids, mainly linoleic, linolenic and conjugated linoleic acids, and decrease in the atherogenic index	[109]
*Saccharomyces cerevisiae*, *Bacillus subtilis* and *Enterococcus faecalis*	5 × 10^11^ CFU/day	Orally, mixed in feed (5 g/day)	Saanen dairy goats (56 days)	Increases the feed intake and milk production and improves the intestinal microbiota	[110]
*E. coli*	10^10^ CFU/calf	Peroral administration (in 200 mL of 10% skim milk)	Weaned calves (32 days)	Reduces the incidence of enterohemorrhagic *E. coli*	[111]
*Lactobacillus casei*, *Lactobacillus lactis* and *Paenibacillus polymyxa*	1.2 × 10^8^ CFU (direct-fed microbial)	Oral	Cattle (84-day fall–winter growing and 140-day spring–summer finishing)	Decreases the spread, frequency and prevalence of *Escherichia coli* O157: H7 in the stool of cattle	[112]
*Saccharomyces cerevisiae* and two strains of rumen-derived *Diutina rugosa*	1 × 10^10^ CFU/mL	Oral administration (100 mL)	Sheep (30 days)	Stabilizes the ruminal pH, improves the richness of rumen microflora, relieves acidosis and inflammation and prevents subacute ruminal acidosis	[117]
*Propionibacterium* P63, *Lactobacillus plantarum* and *Lactobacillus rhamnosus*	1 × 10^11^ CFU/animal/day	Intraruminal cannula (2 g/day)	Sheep (21 days of adaptation and 3 days of challenge)	Stabilizes the pH of the rumen and prevents acidosis	[118]
*Lactobacillus plantarum* strain 220 *Enterococcus faecium* strain 26 and *Clostridium butyricum* strain Miyari	9 × 10^6^ CFU/g 9 × 10^5^ CFU/g 9 × 10^4^ CFU/g	Oral administration	Holstein cattle (14 days of challenge)	Reduces the ruminal pH and the concentration of lactic acid in the ruminal fluid, thus preventing acidosis	[119]

**Table 3 animals-12-00719-t003:** Effects of probiotics on swine production.

Strain(s)	Cell Count	Mode of Administration/Dose	Host/Duration	Effect	Reference
*Lactobacillus plantarum*	5 × 10^11^ CFU/kg	Mixed with the feed	Weaned piglets (28 days)	Improves feed efficiency and decreases rate of diarrhea. Increases serum concentrations of lysine, arginine, serine, glutamate, glycine and alanine, and decreases tyrosine concentration	[127]
*Lactobacillus plantarum* (DSMV 8862) and *L. plantarum* (DSMZ 8866)	Single dose at weaning or suckling of 5 × 10^9^ CFU/mL or 5 × 10^10^ CFU/mL	Oral administration	Piglets (25 and 28 days, respectively)	Improves body weight and feed conversion ratio	[134]
*Lactobacillus plantarum* CAM6	5 × 10^6^ CFU/mL	Oral administration (5 mL)	Weaned piglets (28 days, from 21st to 49th day post-weaning)	Promotes body weight and feed efficiency and increases serum IgA concentration	[135]
*Bacillus licheniformis* and *Bacillus subtilis* spores	1.28 × 10^6^ viable spore/g	Mixed with the feed (400 mg/kg of feed)	Weaned piglets (14 days prior to farrowing)	Improves composition and quality of milk in weaning sows	[138]
*Lactobacillus fermentum* LFQI6	10^10^ CFU/animal	Oral administration	Sows	Decreases weight loss of breeders and mortality of piglets and improves body weight of the litter	[139]
*Lactobacillus plantarum* CAM6	10^9^ CFU/mL	Oral administration (10 mL)	Sows	Improves lactose concentration in milk and decreases diarrheal syndrome and piglet mortality	[79]
*Bacillus subtilis* and its endospores	10^8^ CFU/g	Mixed with the feed	Sows (28 days)	Increases production of milk and plasma proteins and decreases weight loss after lactation	[140]
Probiotic blend		Oral administration (100 mg/kg)	Fattening pigs (12 weeks)	Improves concentration of protein and essential fatty acids in pork, without changes to other meat indicators	[143]
*Pediococcus acidilactici* FT28	2 × 10^9^ CFU/g	Oral administration (200 g/day)	Fattening pigs (28 days)	Improves meat quality with changes to 2-thiobarbituric acid reactive substances, water holding capacity and pH of pork	[91]
*Bacillus subtilis* endospore and *Clostridium butyricum* endospore complex	(*B. subtilis*) 10^10^ viable spores/g (*C. butyricum*) 1.0 × 10^9^ viable spores/g	Mixed with the feed	Growing/finishing pigs (10 weeks)	Increases feed efficiency and meat pigmentation, mainly in redness values and marbling scores	[144]
*Enterococcus faecium*	0.3 × 10^9^ CFU/kg feed	Mixed with the feed	Fattening pigs	Improves meat quality and organoleptic properties such as colorimetry, fat infiltration and meat firmness	[145]
*Lactobacillus plantarum*	1 × 10^9^ CFU/day	Oral administration	Weaned pigs (60 days)	Improves meat pH and various indices of meat texture such as chewiness, restoring force, hardness, stickiness and gumminess	[146]
*Lactobacillus plantarum*	2.2 × 10^8^ CFU/mL	Mixed with the feed (20 mg/kg)	Fattening pigs (42 days)	Change pH, fatty acid and amino acid profile, along with ash, shear force and palatability of the pork	[147]

**Table 4 animals-12-00719-t004:** Effect of probiotics on poultry production.

Strain(s)	Cell Count	Mode of Administration/Dose	Host/Duration	Effect	Reference
*Lactobacillus acidophilus* and a mixture of 12 *Lactobacillus* spp.	0.1% dried culture	Oral, mixed in diet	Broilers	Improves body weight and feed conversion ratio	[163]
*Bacillus subtilis*	8 × 10^5^ CFU/g	Oral, mixed in diet (150 mg/kg)	Broilers	Improves yield traits and increases villus height and villus height/crypt depth ratio in the duodenum	[165]
*Bacillus subtilis*	10^6^ CFU/chick	Oral, mixed in diet	Broilers	Increases villus height and villus/crypt depth ratio in the ileum	[166]
*Enterococcus faecium*	10^9^ CFU/kg of feed	Oral administration with challenge with *E. coli* K88^+^	Broilers	Improves feed efficiency with beneficial changes to intestinal morphology and cecal microflora	[168]
*Lactobacillus acidophilus, Bacillus subtilis, Saccharomyces cerevisiae,*	0.03 mg/kg of feed	Oral, mixed in diet	Broilers	Improves nutrient digestibility, cecal traits and gut morphology	[169]
*Streptococcus faecalis*, *Bifidobacterium bifidum* and *Lactobacillus acidophilus*	10^5^ CFU/mL	Oral inoculation (0.5 mL phosphate-buffered saline (PBS))	Broilers	Improves systemic antibody response to red blood cells	[176]
Probiotics with lactic acid bacteria (*Enterococcus faecium, durans,* *Lactobacillus salivarius* and *E. faecalis*)	1 × 10^7^ CFU/12 g of yeast additives	Oral, mixed in diet	Layinghens	Improves immune response due to increased genetic expression of IL-1β, IL-2 and IFNγ. Increases yolk color and thickness of eggshell. Promotes production of jumbo and extra-large-sized eggs	[177]
*Enterococcus durans*	1 × 10^7^ CFU/feed	Oral inoculation (250 mg/kg of feed)	Broilers	Reduces undetectable levels of *Campylobacter* spp.	[178]
*Lactobacillus acidophilus, Bifidobacterium bifidum* and *Streptococcus faecalis*	Two doses of 1 × 10^5^ and 1 × 10^6^ CFU	Oral inoculation (0.5 mL PBS on day two post-hatch)	Broilers	Decreases expression of IL-12 and IFN-γ and gut protection in broilers challenged with *Salmonella enterica* serovar Typhimurium	[179]
*Lactobacillus acidophilus*, *Lactobacillus casei*, *Enterococcus faecium* and *Bifidobacterium bifidum*	1 × 10^6^ CFU	In ovo inoculation	Broilers	One week after hatching, productivity increases due to the modulation of gene expression in the ileum. Decreases macroscopic lesions and mortality in broilers	[181,183]
*Bacillus subtilis*	1 × 10^7^ CFU	In ovo inoculation (0.5 mL/egg)	Broilers	Stimulates microbial diversity in the GIT and increases amniotic fluid, and in turn, the genetic expression of MUC2 in the ileum	[182]
*Bacillus subtilis*	0.05% dried culture	Oral, mixed in diet	Laying hens	Improves the performance and egg quality	[184]
*Enterococcus faecium*	10 × 10^9^ CFU/g	Oral, mixed in diet	Laying hens	Increases egg production, yolk color and monounsaturated fatty acid profile	[185]
*Bacillus toyonensis* and *Bifidobacterium bifidum*	5 × 10^8^ and 6 × 10^8^ CFU/mL	Oral, mixed in diet	Quails	Improves enzyme activity, productivity, egg quality, fertility and hatchability	[187]
*Saccharomyces cerevisiae*	0.02% dried culture	Oral, mixed in diet	Poultry	Increases body weight and decreases number of goblet cells, although without changes for structure of villi	[193]

## Data Availability

Not applicable.

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
