# Peer review of "Probiotics: Symbiotic Relationship with the Animal Host"

_animals, 2022, doi:10.3390/ani12060719_

Round 1
Reviewer 1 Report
This article could be interesting if some details were added.
It is also necessary to promote reading by adding sub-parts (and sub-titles) and tables.
The article is original with a plan starting with an introduction, then the presentation of in vitro results, the effects of probiotics on the intestinal environment, and their antioxidant capacity. Finally, their effect in different farm animals’ species are presented.
But, as the authors say, lines 423-24, it would be preferable to specify for each study cited the probiotic strain number when possible, or specify when the articles did not mention it.
Likewise, it is very important when the strains are alive, because their metabolism is specific. The article would be considerably improved with these details.
In addition, reading would be easier with more tables and if the sub-parts of each paragraph would be better defined.
From paragraph 3, it is necessary to be very precise on the probiotics’ strain but also on the animal species and the experimental conditions.
Generally, the article lists a lot of work but often does not give enough details for the reader.
In details
- lines 27-28 : « apparently normal » should be changed to “apparently healthy”
- lines 30-32: this sentence is too vague, and unclear. Alive probiotics must resist both acidic pH and basic pH, therefore pH variations, and the bacteriostatic effects of biliary acids and pancreatic juice ...
- lines 35: please define “native” microbiota
- line 49: I don’t understand your choice for the reference [2]. A lot of articles would be more appropriate.
- line 61: when you say “subtherapeutics antibiotics” are still used to reduce mortality, it would be interesting to develop, give an example, and explain the mechanism of action.
- lines 124-125: you must explain the “quorum” concept as soon as this term is mentioned (and not lines 134). Please explain what it is.
- lines 151-152: you are talking about enzymatic hydrolysates so Saccharomyces have an action, even if the yeasts are dead. What is shown with live or dead strains should be easier to understand in the text (maybe classify the studies differently or make subparagraphs, or add a table?)
- line 154: please define co-aggregation.
- lines 160-166: you switch from the specific role of certain B-glucans to the effect of living strains. It's incomprehensible.
- lines 172-173: If you do not present some details of this study, it is not worth quoting it, it does not add anything.
- lines 185-188: please explain and develop the action of probiotics on zono-ocludens and junctional molecule binding…
- lines 328-330: again, the effects on mastitis must be detailed.
- lines 331: the term "poor nutrition" here is too vague, since it is only a situation of acidosis or sub-acidosis.
- lines 339-341: please explain how probiotic can reduce the ruminal ph and preventing acidosis.
- lines 346-348: as you say, the effects of probiotics depend on the concentration, the state (alive or not), the strain ... and for the animal many factors (age, diet ...), also the studies presented must be sufficiently detailed so as not to have to systematically reread the initial article.
- table 1, ref 103: please add details on calves, age and regimen.
- “weaning is associated in pigs with….a high mortality rate”. Please specify the orders of magnitude without probiotic to then understand how the addition of probiotic improves performance.
- lines 378-381: if you do not better explain the observed effects (in whom? when? ....), citing this work is useless.
- lines 406-408: it is not clear ... which addition of which probiotic, in which quantity and when? is it capable of modifying the protein content of meat and essential fatty acids, and at what intensity?
- lines 420-422: idem, please explain with details, otherwise it is useless.
- Lines 423-425: this sentence must guide your work, it must be said well beforehand, and you lead to giving all the information necessary for each study.
- table 2: please be clearer with cell count and final dose for the animal.
- lines 456-459: thank you for being more specific (number ?)
- line 474: idem (villi height)
- line 484: you explain that there is a retrograde peristalsis in rectum, but how can GIT bacteria be a source of essential amino acids?
- line 503: please check if it is TNF-
- lines 539-540: what does mean “Rodriguez (…) using up to 30% torula yeast” ?. If he added 30% of torula yeast in food, you must present also the chemical composition of the regimen.
- table 3: same remark as table 2; what is the final dose for the animal ?
Author Response
Dear reviewer,
Thank you very much for your comments on our manuscripts. We made significant changes in all sections to achieve a better understanding of the manuscript.
This article could be interesting if some details were added.
In details
- Reviewer: lines 27-28: « apparently normal » should be changed to “apparently healthy”
Authors: Done
- Reviewer: lines 30-32: this sentence is too vague, and unclear. Alive probiotics must resist both acidic pH and basic pH, therefore pH variations, and the bacteriostatic effects of biliary acids and pancreatic juice ...
Authors: It is mentioned in a general way the characteristics that a probiotic must have to survive.
- Reviewer: lines 35: please define “native” microbiota
Authors: the native microbiota (microbes which are present in their originated place).
Reviewer: line 49: I don’t understand your choice for the reference [2]. A lot of articles would be more appropriate.
Authors: The reference was replaced by another more appropriate article.
- Reviewer: line 61: when you say “subtherapeutics antibiotics” are still used to reduce mortality, it would be interesting to develop, give an example, and explain the mechanism of action.
Authors: A paragraph was written, which is marked in yellow. L61-66.
- Reviewer: lines 124-125: you must explain the “quorum” concept as soon as this term is mentioned (and not lines 134). Please explain what it is.
Authors: A paragraph was written, which is marked in yellow. L189-192.
- Reviewer: lines 151-152: you are talking about enzymatic hydrolysates so Saccharomyces has an action, even if the yeasts are dead. What is shown with live or dead strains should be easier to understand in the text (maybe classify the studies differently or make subparagraphs, or add a table?)
Authors: Done. L 221-230 and Table 1.
- Reviewer: line 154: please define co-aggregation.
Authors: Done. L221-223.
- Reviewer: lines 160-166: you switch from the specific role of certain B-glucans to the effect of living strains. It's incomprehensible.
Authors: We rewrite the idea for better understanding. L242-45.
- Reviewer: lines 172-173: If you do not present some details of this study, it is not worth quoting it, it does not add anything.
Authors: Done.
- Reviewer: lines 185-188: please explain and develop the action of probiotics on zono-ocludens and junctional molecule binding…
Authors: Done. L262-266
- Reviewer: lines 328-330: again, the effects on mastitis must be detailed.
Authors: Done. A paragraph was written, which is marked in yellow. L405-413.
- Reviewer: lines 331: the term "poor nutrition" here is too vague since it is only a situation of acidosis or sub-acidosis.
Authors: We specify “Diets low in fiber and nutrients”. L419.
- Reviewer: lines 339-341: please explain how probiotic can reduce the ruminal pH and preventing acidosis.
Authors: Done. A paragraph was written, which is marked in yellow. L436-439.
- Reviewer: lines 346-348: as you say, the effects of probiotics depend on the concentration, the state (alive or not), the strain ... and for the animal many factors (age, diet ...), also the studies presented must be sufficiently detailed so as not to have to systematically reread the initial article.
Authors: Done. In the table 2 refers to concentration, dose and duration of treatment.
- Reviewer: table 1, ref 103: please add details on calves, age and regimen.
Authors: Done. Added to all references, now is Table 2
- Reviewer: lines 378-381: if you do not better explain the observed effects (in whom? when? ....), citing this work is useless.
Authors: Done. A paragraph was written, which is marked in yellow. L472-479.
- Reviewer: lines 406-408: it is not clear ... which addition of which probiotic, in which quantity and when? is it capable of modifying the protein content of meat and essential fatty acids, and at what intensity?
Authors: Done. A paragraph was written, which is marked in yellow. L503-510.
- Reviewer: lines 420-422: idem, please explain with details, otherwise it is useless.
Authors: Done. A paragraph was written, which is marked in yellow. L517-527.
- Reviewer: Lines 423-425: this sentence must guide your work, it must be said well beforehand, and you lead to giving all the information necessary for each study.
Authors: Done.
- Reviewer: table 2: please be clearer with cell count and final dose for the animal.
Authors: Done, now is Table 3.
- Reviewer: lines 456-459: thank you for being more specific (number ?)
Authors: above 10 log CFU/g
- Reviewer: line 474: idem (villi height)
Authors: Done. A paragraph was written, which is marked in yellow. L575-580.
- Reviewer: line 484: you explain that there is a retrograde peristalsis in rectum, but how can GIT bacteria be a source of essential amino acids?
Authors: Done. A paragraph was written, which is marked in yellow. L591-594.
- Reviewer: line 503: please check if it is TNF-
Authors: Done.
- Reviewer: Lines 539-540: what does mean “Rodriguez (…) using up to 30% torula yeast”? If he added 30% of torula yeast in food, you must also present the chemical composition of the regimen.
Authors: Done. A paragraph was written, which is marked in yellow. L649-658.
- Reviewer: table 3: same remark as table 2; what is the final dose for the animal?
Authors: Done. Now is Table 4.
Reviewer 2 Report
Comments to Author
The work from Melara and colleagues demonstrated Probiotics: Symbiotic relationship with the animal host. The manuscript is generally well written. However, I would like to raise the following points for the authors' consideration.
- Please correct grammatical issues throughout.
- Introduction: add a brief introduction about probiotics in general at the very beginning of the manuscript including their pros and cons.
- Material and Methods: this section is missing from the manuscript. The authors should add this section as section “2” of the manuscript, providing information related to their literature search: which data bases were used, which keywords, time range etc.
- I would strongly recommend to include the limitation of the literature review on Probiotics in the discussion.
- What research are necessary in future for improvement in the health services?
Author Response
Dear Reviewer,
Thank you very much for your comments on our manuscripts. I have made substantial changes to achieve a better understanding of the manuscript.
Comments and Suggestions for Authors
Reviewer. The work from Melara and colleagues demonstrated Probiotics: Symbiotic relationship with the animal host. The manuscript is generally well written. However, I would like to raise the following points for the authors' consideration. Please correct grammatical issues throughout.
- Reviewer: Please correct grammatical issues throughout.
Authors: Done.
- Introduction: add a brief introduction about probiotics in general at the very beginning of the manuscript including their pros and cons.
Authors: Done. We write a paragraph about your suggestion, which is marked in yellow. L 81-90.
- Material and Methods: this section is missing from the manuscript. The authors should add this section as section “2” of the manuscript, providing information related to their literature search: which data bases were used, which keywords, time range etc.
Authors: Done. A detailed description of the materials and methods was made, which are marked in yellow. L 139-175.
- I would strongly recommend including the limitation of the literature review on Probiotics in the discussion.
Authors: Done.
- Reviewer: What research are necessary in future for improvement in the health services?
Authors: Done. We write a paragraph considering your suggestions, which is marked in yellow. L 125-133.
Round 2
Reviewer 1 Report
see the file

Author Response
Dear reviewer,
Thank you very much for your comments on our manuscript
The article was deeply improved.
I still have a few formal remarks and two substantive remarks.
Formal remarks:
- Reviewer: Line 5: is it useful to write “Master in sustainable tropical agriculture”?
Authors: Done. The affiliation was corrected as stipulated by the University. “Sustainable Tropical Agriculture Master Program, Zamorano University”
- Reviewer: Lines 81-84: after this sentence, we expect to find references
Authors: Done.
- Reviewer: Page 5 in the table 1: there is still an asterisk after Wickerhamomyces anomalus
Authors: Done.
- Reviewer: Line 173 (is the line 473): “a single dose at birth and lactation” is unclear
Authors: Done. We specify the details of the doses “A study in pigs showed that a single oral dose at 5 x 1010 CFU/mL or 5 x 109 CFU/mL in suckling piglets or at weaning with L. plantarum (DSMV 8862) and L. plantarum (DSMZ 8866), respectively”
- 5. Reviewer: Lines 526 and 527: what do (a*) and (b*) mean?
Authors: We clarified the meaning of the abbreviations defined by the International Commission on Illumination "yellowness (b*) and redness (a*)
- Reviewer: Table 3: add a space between 5 and ml + thank you for being homogeneous in the notation of the units (example CFU/g or CFU.g-1). This last remark applies to the
entire text and units should be expressed as recommended by the journal.
Authors: Done.
- Reviewer: Line 651: B12 would be replaced by B12
Authors: Done
- Reviewer: Line 652: no space before 1.20%
Authors: Done.
- Line 654: no space after 22 in (22 -42 days)
Authors: Done.
Substantive remarks:
Reviewer: Lines 419-422: This sentence is false. I propose “To meet the high nutritional needs of cows during lactation, diets are often low in physically effective dietary fiber and rich in concentrates, leading to rumen acidosis.”
Authors: Done.
- Reviewer: Lines 423-427 in the first version, now lines 527-530: I would have preferred that you move this sentence to the introduction because it concerns all of the work. One of the limits of your synthesis is that, in fact, all the papers do not clearly mention the microbial strain used.
Authors: Done. L. 116 – 119.
Reviewer 2 Report
- Methodology: Sources and keywords are not clearly mentioned.
- Where is the limitation of this review on Probiotics that should be in the discussion?
- I would suggest to move the sentences: Line125-133 to the end of the discussion section.
Author Response
Reviewer 2
Thank you very much for the comments to improve our manuscript
Reviewer: Methodology: Sources and keywords are not clearly mentioned.
Authors: Done. We detail the sources and keywords for conducting the review.
- Reviewer: Where is the limitation of this review on Probiotics that should be in the discussion?
Authors: Done. We consider other papers published in Animals.
- Reviewer: I would suggest to move the sentences: Line125-133 to the end of the discussion section.
Authors: Done.